# Health Literacy: From a Property of Individuals to One of Communities

**DOI:** 10.3390/ijerph17051601

**Published:** 2020-03-02

**Authors:** Candan Kendir, Eric Breton

**Affiliations:** 1École des hautes études en sante publique (EHESP), 35043 Rennes, France; eric.breton@ehesp.fr; 2Laboratory Arènes (UMR CNRS 6051), 35700 Rennes, France

**Keywords:** health literacy, social determinants of health, empowerment, community health

## Abstract

Health literacy (HL) is increasingly hailed as a strategy to improve the control individuals have over their health. A central critic of HL intervention is its overemphasis on individual level factors, something recognised in the 2008 report of the Commission of Social Determinants of Health (SDoH) that recommended expanding the scope of HL to cover the SDoH. The objective of our study was to assess the extent to which recent progress on HL captures the need for collective action on the SDoH. We conducted a scoping review on PubMed looking for review papers published between 2013–2018 in English and French. Definitions of HL were analysed against two main dimensions (i.e., locus of change of HL strategies and foreseen outcome of HL improvements). Despite a number of authors calling for more research on HL interventions at the community level and an expansion of the definition to cover the SDoH, we found that the recommendation of the Commission has yet to be implemented. Even when the definitions include the capacities of individuals on distal determinants, both the locus of change and outcomes of HL improvement do not go beyond intra individual factors (knowledge, skills, etc.). It is noteworthy that communities were either framed as a setting outside of health care services or as an aggregate of individuals. We found no instance of HL intervention regarding communities as complex systems of actors sharing a common space and dynamic. We conclude by suggesting a new definition of HL and by drawing attention to the research gap in addressing the upstream SDoH through HL actions.

## 1. Introduction

A decade has passed since the Commission of Social Determinants of Health (SDoH) published its milestone report with, at its core, a call to improve population’s daily living conditions as these are at the root of health inequalities between and within countries [1]. Included in its recommendations, albeit greatly overlooked, was a plea to expand the scope of health literacy (HL) to reflect the agenda on the SDoH [1]. This recommendation came as no surprise considering the potential benefits HL improvement strategies bear in empowering people and increasing the control the population has on its own health [2]. Today, the contribution of HL, in preserving and promoting people’s health, has it ranking as a SDoH in itself [3]. Yet we know little on how HL as a research and practice field integrates this recommendation. Unknown as well is how HL can integrate the fact that work on the SDoH cannot be subsumed to an individual enterprise. Little progress can be expected on the upstream determinants of health such as the ones in relation to the quality of housing, jobs, urban settings, leisure activities, and transport systems without a real and sustained collective effort. This calls for investigating whether being health literate can also be considered a property of communities or groups and not just of individuals [4]. In this paper, we report on the results of a scoping review exploring the way HL reflects the need for action on the SDoH.

## 2. Materials and Methods

Two general questions guided our scoping review: How is the scientific literature linking HL with the upstream SDoH? How does HL reflect the need for collective action on these SDoH?

To answer our research questions, we conducted a scoping review from March to July 2018. We searched the PubMed database with the term “health literacy”. We included all the systematic reviews and review papers that were published between January 2013–January 2018 in English and French that feature a definition of HL (see Figure 1 for the inclusion/exclusion criteria). We analysed the papers against four main dimensions: (1) the definitions of HL they featured, (2) their linkage to the SDoH, (3) the target of actions, i.e., whether the actions were on individuals or communities and, (4) the nature of the determinants of health targeted.

We felt it was important to not just distinguish between actions on the intra-individual determinants vs. actions on upstream determinants. We hypothesized that a programme aimed at changing intra-individual factors (e.g., knowledge, attitudes, skills, motivation) could still prove to be thought out as a strategy to generate changes in the upstream SDoH (e.g., improving knowledge on health services to increase capacity to advocate for system change).

The analysis was also carried out on relevant HL documents available on the WHO websites. In addition, the review of the definitions of HL conducted by Sørensen et al. [3] has proven a valuable starting point to appraise the range of definitions available that could not be captured within the time period covered by the search of databases. In order to get a better history of the evolution of this strategy and identify definitions from the past, we also completed our dataset of papers by scanning the reference lists of the articles retrieved from PubMed. Additional searches on PubMed and Google Scholar were also carried out to answer specific questions that arose during the analysis.

## 3. Results

Since its inception in the 1970s, there has been a growing interest in the concept and methods of HL, with numerous papers and guidelines published in medical and allied health sciences journals [5]. This activity has yielded a number of significant improvements both in terms of its conceptualisation and in better appraising its contribution to the reduction of inequalities in health by empowering individuals to gain better control over their health and wellbeing [6].

As a concept and a set of practices and methods, HL emerged from two very different roots; one being from medical care with its patient education focus and the other one from the broader public health field with its health promotion perspective [7]. It is a strategy that addresses not only the shortfalls of the individuals in terms of skills and knowledge but also the deficits health services may display in answering the needs of the population (for instance by increasing the readability of consent forms and improving communication skills of health care providers) [8,9,10]. This dual purpose lead to numerous investigations to better capture the scope of HL and also to devise better strategies to improve people’s HL, particularly in the clinical context [11,12,13,14,15,16,17]. Adolescent programs to improve food literacy [18], patient education programs [12], vaccination [19], and mobile applications to improve alcohol HL [20] are only some of them.

Our search generated 173 articles from which we retrieved nine definitions. See Appendix A for the whole set of definitions. We summarised a selection of definitions in Table 1 that illustrates the evolution of HL. Of those definitions, Ratzan and Parker was cited 112 times, Nutbeam 29 times, Sørensen et al. 22 times and Kickbusch three times. Freedman et al.’s definition was not cited in the papers that we retrieved from literature, it came out of the Sørensen’s et al. study. In the papers we retrieved from the literature there was also a definition by Canadian authorities (cited six times), by Australian authorities (cited one time) and the definition of Berkman et al. (three times).

Simonds is largely credited for first alluding to HL. Although he did not formally define the concept, we can infer from his paper that it is an instrument enabling individuals to benefit from the progress brought about by biomedicine and from the knowledge on harmful and health-promoting behaviours. Acknowledging that much happens in terms of choices and behaviours before the unset of diseases, he defends the delivery of better-quality health education at schools by qualified teachers [5]. Following this first sketching of the idea of HL, the definitions gradually expanded to encompass the idea of an enhanced understanding and informed decision-making on health information and learning to find one’s way in the health care system [22,25].

Nutbeam’s definition published in *Health promotion International* [26] is the journal version of his WHO-published *Health promotion glossary* [21]. In the accompanying rational to his definition, he recognised that the outcome of HL is not only healthy individuals but also a healthy community. Furthermore, he acknowledged the role HL can fulfil in improving living conditions which brings him close to the views of the Commission of the SDoH.

It is difficult to say where Ratzan and Parker’s definition of HL comes from. It is briefly presented in the introduction of their National Library of Medicine bibliography, which features 479 citations [22]. Readers are left on their own to appraise the scope of this definition.

Kickbush et al.’s [23] definition was published in an International Longevity Centre-UK commissioned report. It is markedly limited to changes at the individual level with no reference to the communities nor to the upstream determinants of health.

Freedman et al.’s [24] contribution falls well outside the area inhabited by previous definitions. Published in 2009, theirs was contemporary to the work of the Commission of the SDoH in which they obviously tap into as is evidenced by their reference to it in the text. They clearly position HL in the issue of addressing the SDoH.


*“These individual-level health literacy initiatives may do very little to achieve the ultimate goal of promoting equitable health status because “they do not address the root causes of health illiteracy, such as socioeconomic disparities and unequal access to high quality education.”*
[24] (p. 447)

Freedman et al.’s answer to this shortfall of HL was proposing to broaden its scope under what they refer to as *public health literacy*.

Sørensen et al. [3] crafted the latest much-cited definition we found. Their contribution is an outcome of the HLS-EU collaborative European project and was fed by the analysis of 17 definitions published in peer-reviewed papers (of which three are listed in Table 1). Theirs includes the same three core dimensions (i.e., access, understand and apply health information) found in Nutbeam [26], Ratzan and Parker [22] and others such as the one of the WHO European office Health 2020 strategy: “*the cognitive and social skills which determine the motivation and ability of individuals to gain access to understand and use information in ways which promote and maintain good health*”. Yet Sørensen’s et al. includes, as Freedman et al.’s does, a fourth one to “appraise information” that reflects a capacity for critical thinking.

Another interesting feature of the definition of HL defended by Sørensen et al. is the much broader scope it embraces to include Freedman’s public HL. Applying a dynamic life-course perspective, they identified three domains in which individuals can apply their HL skills: health care, disease prevention and health promotion. Whereas the health care domain is about medical information and the disease prevention one is about risk factors; the health promotion domain is the one in which information on the SDoH are actually addressed. According to Sørensen et al., dimension of HL in the domain of health promotion includes (1) ability to regularly update oneself, (2) comprehend information, (3) interpret and critically think and (4) ability to make informed decisions on determinants of health in the social and physical environment [3].

## 4. Discussion

After reviewing the most authoritative definitions of HL, we found that a common feature of the contributions to research and practice is that whether one is interested in the relationship between HL and certain health-related behaviours such as disease management, medication adherence, screening, and patient-provider communication or interested in interventions to improve the level of HL, the main dominant focus remains on intra-individual determinants of health [27,28,29,30,31,32]. Our study brings a unique contribution as we found no other paper analysing HL programmes against the category of determinants targeted.

This focus on intra-individual determinants is at odd with mounting evidence on the weight of upstream SDoH in explaining population health statuses. There is now a long history of works demonstrating the relatively modest impact health care services and health-related behaviours have on population health (see for instance the pioneering works of McKeown on the contribution of medicine [33]). Improving access to a nutrient-rich diet, a proper education system, good housing and working conditions, water and sanitation, as well as supportive social environments and reducing discrimination based on gender identity, race and religion can have far more impact on population and individual health than solely focusing on health care services and health-related behaviours [34]. This is to say that a person holding an adequate level of HL may be much more capable of understanding, interpreting information and critically thinking and making informed decisions on health-related matters yet be seriously impeded in her capacity to take control over her health due to daily living conditions related to structural/environmental determinants (e.g., insufficient income, lack of transportation or safe outdoor spaces). In this case, even if services are HL responsive, there will be a limit to what an individual can achieve for health improvement.

For this reason, addressing HL as a collective property is essential. Upstream determinants of health are powerful drivers of inequities, so much so that one can observe a social gradient in the level of HL [2]. Capacity for HL being, for structural reasons, ill-distributed in societies implies that not only: (1) population groups at the bottom of the social ladder are at risk of showing poor HL level, but (2) that at the individual level, poor HL is a contributing factor to health inequalities in the society through bad command over one’s health- and wellbeing-related daily decisions. It is therefore critical that efforts to increase HL across the whole social spectrum are also compounded by actions to increase the level of community HL to address the structural determinants of health. For this reason, one can legitimately claim that HL interventions that do not go beyond enhancing people’s agency (HL seen only as a property of individuals) are falling short of engaging the full potential of this strategy. Yet, there are still few papers reporting on interventions targeting the whole community and most do not explicitly mention impacts at the community level [35,36].

More upsetting still is the recognition that actions aiming at HL improvement only as a property of individuals may potentially increase health inequities within populations. We see two reasons for that. First, due to what is referred to as the “Inverse information law” [4,37]; i.e., people with low HL level having in the first place less access to health information and the areas to improve HL. Second, in order to be converted into health promoting behaviours and outcomes, information and skills also require a whole range of other resources (e.g., money, social support, transport, infrastructure, security, entitlements). A number of theoretical contributions point to the ill distribution of these resources across social classes to explain health inequities [38].

However, despite this prevailing focus on intra-individual determinants of health as the locus of change, we saw some progress in embracing the agenda of the SDoH [3,24] and calls have been issued for moving HL from an individual to community or population level [24,39]. Of much relevance to the point we defend is the identification by Zarcadoolas and colleagues [40] of “civic health literacy” as a core dimension of HL. In their own words, civic HL: “*refers to abilities that enable citizens to become aware of public issues and to become involved in the decision-making process*”. This dimension of HL includes: (1) media literacy skills, (2) knowledge of civic and governmental processes and, (3) an awareness that individual health decisions can impact public health.

### 4.1. Towards a New Concept for Community Health Literacy

We believe that in order to have effective community HL interventions, definitions and measurements should also encompass the community level. However, for this to happen we need a clear distinction of HL as a property of individuals and HL as a property of communities.

Two important aspects need to be raised at this point. First of all, not addressing the distal factors leaves unbridled the negative feedback loop of an individual with low HL living in a community also deprived of HL. Second, as the health of a community cannot be subsumed to the mere aggregation of individual risk factors and health statuses, collective action on population health will eventually affect its member’s health and wellbeing. In Table 2 we summarize the contrast between what we regard as two general perspectives on HL as informed by the agenda of the SDoH. In light of this, while improving HL of the individuals addresses the proximal determinants and HL-responsive services address some of the barriers to converting care into better health and wellbeing, it mostly leaves distal ones untouched.

Recognising that a community, seen as a complex system of actors, is more than the sum of its individuals, one may also question whether a community considered to be displaying a high level of HL on the basis of the sum of its health literate individuals is indeed a health literate community. In other words, is HL only a property of individuals or can the concept/strategy be expanded to make it also a property of a community? This is a topical issue as again many powerful determinants of health require collective action to be addressed.

This issue has somewhat been reflected over the years in the literature. Kickbusch [41] argued that health promotion research and intervention require a much stronger focus on communities than on the individuals, considering the dynamic interaction between the SDoH. The role of improved HL in community health was also addressed in one WHO factsheet [2]. In it, HL is considered an asset for the communities contributing to their resilience and playing an active role in promoting health when combined with the appropriate social resources such as self-help groups and neighbourhood support. However, and regardless of whether the targets of HL interventions are individuals or communities, most of the scientific production on the topic focuses on changes at the individual level rather than at the community [35]. A noteworthy outlier in this literature is the contribution of Freedman et al. who, following the work of the CSDH, advocated to set HL also as a collective level enterprise to address the SDoH [24]. However, from the standpoint of existing HL interventions targeting communities we found in the literature, the community is either understood as a setting (a place outside of the medical clinics) or as an aggregate of individuals [35,36]. Similarly to De Wit et al. [42], we found that the focus of these studies was mainly on vulnerable populations such as migrant groups [43] and older people [35].

### 4.2. The Way Forward

Despite the fact that HL, as an area for action on the SDoH, was discussed a number of times by different authors [7,24,35,41], we still don’t witness in the scientific literature an expansion of the concept as a property of communities. We therefore tentatively propose the following points to be included in the definition of community HL. In line with the concept of civic literacy [40], a health literate community should be able to identify the problems of its people, mobilise itself to empower its members, use its own assets and seek help related to the defined problem in order to influence decision-making on policies and resource allocations.

We propose the following definition of HL that reflects the need for collective action on the SDoH. We freely and shamelessly borrow from Sørensen et al.’s definition:


*Health literacy is linked to literacy and is both a property/quality of a person and of a community. A person’s health literacy entails her/his knowledge, motivation and competences to access, understand, appraise, and apply information on health and upstream determinants of health in order to make judgments and take decisions in everyday life concerning healthcare, disease prevention and health promotion to maintain or improve quality of life during the life course. When a community is health literate it refers to its capacity to gather information on its upstream SDoH, to mobilize the collective resources to act upon these, to advocate efficiently for structural changes in order to improve the daily living conditions of its members.*


If we accept the idea that being health literate can also be a property of a community capable of addressing the upstream determinants of its health and wellbeing, then the logical question that ensues is what are the levers for improved community HL on the SDoH? Could, to a certain extent, health literate communities overcome through community mobilisation their state of disfranchisement and, for instance, the structural and institutional racism they experience? Before being able to provide elements of answer, efforts and resources in research should be invested in identifying the assets a community needs to mobilise for collective action on the structural/environmental determinants of health. Obviously collective action requires first of all recognizing the structural determinants of population health. This could be followed by determining the necessary steps forward such as increasing capacity for action, creation of citizen networks or advocating for structural/environmental change.

Although we are confident in the validity of the observations and conclusions we derived from our dataset, our study is not exempt of limitations. Constraints in time and budget had us limit our search of papers to the PubMed database. Whereas we cannot dismiss the possibility of having failed to catch significant contributions, we believe this is unlikely as we looked for systematic review papers and review papers that in most cases carried out searches from multiple databases. We also had to restrict the publication period covered, a limit to our results which was to some extent offset by scrolling through the list of references of the papers we retrieved.

## 5. Conclusions

HL as a research field has yet to embrace the agenda of the WHO Commission on the SDoH by recognising the weight of the upstream determinants of health. However, there is much more to do than simply increasing people’s awareness and knowledge on the SDoH. The field needs to break up its chains from a long dominant individualistic bio-medical perspective on health. As we argued here, not only should individuals be health literate, communities should too.

## Figures and Tables

**Figure 1 ijerph-17-01601-f001:**
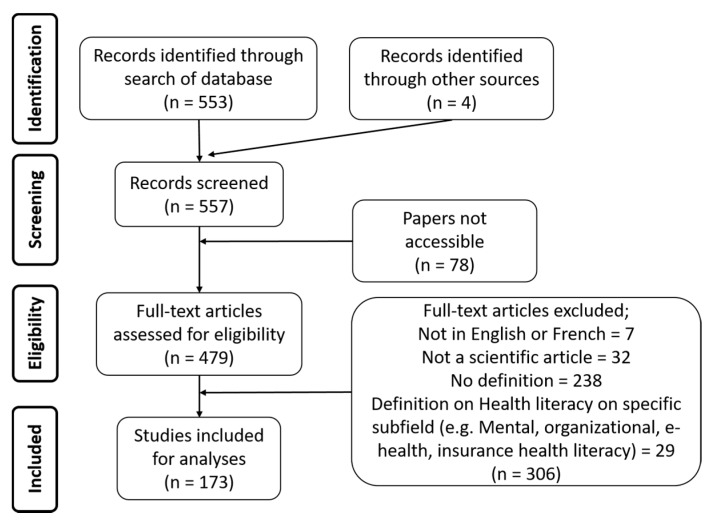
Prisma flow chart.

**Table 1 ijerph-17-01601-t001:** From patient education to health literacy: a selection of definitions quoted in the scientific literature.

Papers Listed from the Oldest to the Most Recent	Definition/Mention	Locus of Change/*Strategies for Change (When Mentioned)*	Outcome of Improved Level of HL
Nutbeam (1998) [21]	“Health literacy represents the cognitive and social skills which determine the motivation and ability of individuals to gain access to, understand and use information in ways which promote and maintain good health.”	“knowledge, personal skills and confidence to take action” *Through education and empowerment*	Change of personal lifestyle and living conditions
Ratzan and Parker (2000) [22]	“The degree to which individuals have the capacity to obtain, process and understand basic health information and services needed to make appropriate health decisions.”	The capacity of individuals to obtain, process and understand basic health information. *Through education*	Better reading and understanding medication labels, appointment slips and other health related materials
Kickbusch, Wait & Maag (2005) [23]	“Health Literacy is the ability to make sound health decisions in the context of everyday life—at home, in the community, at the workplace, in the health care system, the market place and the political arena. It is a critical empowerment strategy to increase people’s control over their health, their ability to seek out information and their ability to take responsibility.”	Skills to make health-related decisions	Increased skills of an individual on one’s own health. Empowered individuals
Freedman et al. (2009) [24]	“Public health literacy is defined here as the degree to which individuals and groups can obtain, process, understand, evaluate, and act upon information needed to make public health decisions that benefit the community.”	Power to organize activities to accomplish public health goals and objectives *through civic engagement*.	Increased skills of individuals/groups to advance the health of communities.
Sørensen et al. (2012) [3]	“Health literacy is linked to literacy and entails people’s knowledge, motivation and competences to access, understand, appraise, and apply health information in order to make judgments and take decisions in everyday life concerning healthcare, disease prevention and health promotion to maintain or improve quality of life during the life course.”	People’s knowledge, motivation and competences	Maintain and improve the individuals’ quality of life during the life course

**Table 2 ijerph-17-01601-t002:** Synthesis of two general perspectives of health literacy.

	Locus of Change for HL Enhancement	Outcome of HL Enhancement	SDoH Addressed
HL as an individual property	Understand, interpret and critically analyse health information. Make informed decision on health and wellbeing	Improved communication in health, improved health-related behaviours, increased adherence to medications, increased individual participation to health promotion and prevention activities, decreased morbidity and mortality	*Proximal factors* Individual lifestyle factors, Biological factors, Behaviours
HL as a collective property	Knowledge on the broader, upstream determinants of health. Capacity for community mobilisation for change. Capacity for policy advocacy	Change in upstream SDoH, improvement in daily living conditions (housing, social support, urban design, etc.)	*Distal factors**(that may have a significant impact on proximal ones)* Environmental factors, Living conditions

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
