# Peer review of "Health Literacy: From a Property of Individuals to One of Communities"

_ijerph, 2020, doi:10.3390/ijerph17051601_

Round 1
Reviewer 1 Report
Thank you for addressing my issues.
Reviewer 2 Report
The Authors have improved the paper according to the suggestion
This manuscript is a resubmission of an earlier submission. The following is a list of the peer review reports and author responses from that submission.
Round 1
Reviewer 1 Report
Your proposed article is timely. You build the case for exploring health literacy from a broader perspective, more specifically exploring the public health relationship of health literacy and social determinants of health. Using health literacy as a collective/community concept e.g. health literate community is in line with some of the research I am exploring. Although I enjoyed reading this article, it needs to be revised to address certain language, grammar, punctuation and format issues.
You build on the Stevenson definition which has evolved from previous other definitions. I noticed that Christina Zarcadoolas definition of civic literacy was not considered.
You put forth a point of view concerning the ability of community being able to initiate change from within; however, you have not addressed the impact of structural and institutional racism as a barrier for communities becoming health literate and initiating collective action to improve the health status of not only the individual but also the community.
Reviewer 2 Report
Thank you for offering opportunity to review this paper. This paper has addressed an important issue in health literacy, that is, lack of understanding of the application of health literacy at community level.
However, authors are recommended to consider this paper as ‘discussion paper’ rather than ‘a scoping review’. Scoping review should be developed with its standardized structure. This paper (https://www.ncbi.nlm.nih.gov/pmc/articles/PMC4491356/) would be helpful to the authors to reconsider the nature of this paper.
Methods: More detail description of the extraction of information from the reviewed articles is needed. Who have involved in this review process? Is there any checklist or strategy to extract relevant information from the reviewed articles? How does the project team determine which articles should be excluded or included? Criteria for including or excluding the papers should be stated clearly so that the readers can know whether the search is comprehensive. How many articles are finally included in this scoping review? Analysis plan could be given in details. Other than retrieving and reviewing the definitions of health literacy and the focus of actions are at individual level or community level, what else did you investigate? Did you go in-depth investigation for the service recipients’ responses to the interventions, if any? Some interventions target for individuals but it has impact on community health and decisions made by community members/families/policy makers. If such a situation, are you counting this health literacy intervention as at individual levels? Not community level?
Results: from Line 66 to 68, the authors consider that there is ‘dual purpose’ of health literacy interventions which serves to fulfill individuals’ lack of skills and knowledge, as well as the deficit health service providers may display. I think this idea is worthy to be elaborated. Evidence (may be from the reviewed articles) should be given to illustrate this idea, and hypothesize the possible consequences of this dual purpose is not fulfilled, as planned. Examples in Line 68-70 should be supported with further explanation how these programmes support dual purpose.
Among the 645 articles searched, how many articles contributed to the six central definitions of health literacy? Other than the six articles included in Table 1, what other articles have cited these definitions and acknowledged these definitions as crucial to the conceptualization of health literacy? Additional information or examples would be useful to explain which definition(s) are illustrating the inclusion of community level of health literacy.
Discussion: The first paragraph intends to state the current level of health literacy remains to be at ‘individual level’ focusing on intra-individual determinant of health. Such argument needs amber evidences and I think it is not difficult to get all these evidences from the existing literatures. Case studies or key findings of the reviewed articles could be used as the evidences. Health-related behavours could be either at individual level or community level. Cautions should be taken to cite the appropriate examples here.
Definition and examples of upstream social determinants of health (SDoH) could be provided to make readers understand better the argument in Paragraph 2. Some concrete examples (from the reviewed articles) to show how health literacy is connected to health decision making despite the existence of unfavorable physical environment or economical insufficiency persists – what did government /health professionals help and support individuals? How individuals improve own health literacy or make health decision? Etc. The present format seems to advocate for redefine health literacy without illustrating the experiences / evidences of such trial as shown in the literatures/previous research studies.
Please elaborate the idea from the sentence (Line 153-154) ‘…HL improvement only as property of individuals may potentially increase health inequalities among individuals.’ This idea justifies well about the application of health literacy in community level, so further explanation or illustration would be helpful to the readers to understand the advocacy.
Line 158 mentions about the interventions that moved to community or population level are limited. Please state what was done in these interventions and how the community responded to these interventions. As stated in your objectives, you wish to highlight the recent progress on health literacy and assess how these captures the need for collective action on the SDoH.
Minor editing: a few lines show ‘In Error! Reference source not found’. Please check.
Reviewer 3 Report
The paper describes the results of a scoping review. The aim of the study is not clear and the research question is not reported. Moreover, I have many concerns about the methodological approach. In fact, the authors stated they have conducted a scoping review but the methods they used seems those of narrative reviews. I suggest to refer to some key articles defining literature reviews, for example, Munn, et al., BMC Medical Research Methodology, 2018; 18:143; 19; Peters et al., Int J Evid Based Health 2015, 13:141–6.
The results and the discussion seem partially overlapped: I suggest to better split up the two sections, describing in the results what emerged from the selected papers and arguing the results in the discussion.
Despite these critical issues, the discussion presented some interesting ideas for conceptualization and for future researches
Introduction
This section is a bit confusing and reductive: the authors have to argued more on the relationship between health literacy and social determinant of health, as well as on why health literacy is nowadays considered as a social determinant of health itself. The aim is unclear: it needs to be specified, detailing the research query. What do the authors mean by “HL reflects the need for action on the SDoH”? Moreover, the aim seems to be not in line with the titleMaterials and methods
Two databases (PubMed and Google Scholar) are too few for a scoping review. Other databases have to be added, for exemple Embase, Web of Science, Health Evidence The search strategy in unclear, so the scoping review is not transparent and reproducible. Specifically, inclusion and exclusion criteria are not described, it is not specified whether only one author conducted the selection process or multiple reviewers were involved in order to reduce error and increase reliability Methods used for data analysis and synthesis are not described Line 47: what do the authors mean for “reference”? it is not clear Lines 53-55: the authors have to describe the additional searches strategiesResults
Lines 57-59: I suppose this is not a result of the study, since the reference 1 and 2 are out of the data range for selecting papers Lines 62-64: idem The scoping reviews, as well as the systematic reviews, generally start with the number of papers included in the final synthesis. Lines 72-73: there is a problem with a reference Table 1: why do the authors are interested in describing the “outcome of improved level of HL”? it is not clear, neither from the aim of the study, nor from the methods sectionDiscussion
Numerous interesting ideas are described, with innovative aspects, which however do not emerge clearly from the results obtained In my opinion, Table 1 (that should be table 2, since table 1 is in the results section) could be placed in the results section, as a synthesis of the analysis of the selected papers
